# Characterisation and chemometric evaluation of 17 elements in ten seaweed species from Greenland

Katharina J. Kreissig[1], Lisbeth Truelstrup Hansen[1], Pernille Erland Jensen[2], Susse Wegeberg[3], Ole Geertz-Hansen[4], Jens J. Sloth[1]*

1 National Food Institute, Technical University of Denmark, Kgs. Lyngby, Denmark, 2 Department of Civil Engineering, Technical University of Denmark, Kgs. Lyngby, Denmark, 3 DCE - Danish Centre for Environment and Energy / Department of Bioscience, Aarhus University, Roskilde, Denmark, 4 Greenland Institute of Natural Resources, Nuuk, Greenland

* jjsl@food.dtu.dk

**Data Availability Statement:** The data underlying the results are available from DTU Data, the research database of the Technical University of Denmark (doi.org/10.11583/DTU.13251575).

## Abstract

Several Greenland seaweed species have potential as foods or food ingredients, both for local consumption and export. However, knowledge regarding their content of beneficial and deleterious elements on a species specific and geographical basis is lacking. This study investigated the content of 17 elements (As, Ca, Cd, Cr, Cu, Fe, Hg, I, K, Mg, Mn, Na, Ni, P, Pb, Se and Zn) in 77 samples of ten species (*Agarum clathratum*, *Alaria esculenta*, *Ascophyllum nodosum*, *Fucus distichus*, *Fucus vesiculosus*, *Hedophyllum nigripes*, *Laminaria solidungula*, *Palmaria palmata*, *Saccharina latissima* and *Saccharina longicruris*). Element profiles differed between species but showed similar patterns within the same family. For five species, different thallus parts were investigated separately, and showed different element profiles. A geographic origin comparison of *Fucus* species indicated regional differences. The seaweeds investigated were especially good sources of macrominerals (K > Na > Ca > Mg) and trace minerals, such as Fe. Iodine contents were high, especially in macroalgae of the family Laminariaceae. None of the samples exceeded the EU maximum levels for Cd, Hg or Pb, but some exceeded the stricter French regulations, especially for Cd and I. In conclusion, these ten species are promising food items.

## Introduction

Marine macroalgae, commonly known as seaweeds, are increasingly becoming popular as food items in the Nordic countries [1], as well as in Greenland [2, 3], where they have been a part of the traditional Inuit diet [4, 5]. Moreover, seaweeds have been identified as a sustainable income source in the remote and sparsely populated areas of the Northern Periphery and Arctic region of Northern Europe and Greenland—a region with a low population density and pristine waters [6].

Having detailed insight in the nutritional composition of macronutrients (lipids, carbohydrates, proteins, etc.) and minor components, including essential and non-essential elements,

**Funding:** KJK was funded by Pinngortitaleriffik case number 80.24. The funders had no role in study design, data collection and analysis, decision to publish, or preparation of the manuscript.

**Competing interests:** The authors have declared that no competing interests exist.

is important for both currently consumed seaweed species and species of interest for future human consumption [7].

Seaweeds have a highly variable nutritional composition [8, 9] but are generally good sources of minerals and iodine [8, 10]. However, in some cases they are also known to contain undesirably high concentrations of certain chemical elements, which have been identified as hazardous: As, Cd, Hg, I and Pb [11–14]. This is attributed to the accumulation of cations from the seawater through their association with biopolymers [15] or, in the case of iodine in some species, its function as an antioxidant [16, 17].

The contents of elements of concern (such as As, Cd, Hg, I, Ni, Pb) need to be mapped for the individual seaweed species. Currently, there is limited European legislation on maximum levels allowed in seaweeds, with stricter regulations on a national level found only in France. Meanwhile, to assess the dietary exposure of the population through the consumption of seaweeds, the EU is collecting information on the occurrence of As, Cd, Hg, I and Pb in a range of seaweeds and products based on seaweeds during the period from 2018 to 2020 [18].

While the nutritional composition of Nordic seaweeds has been studied intensely in recent years, and is increasingly well described [8, 9], there is a distinct lack of knowledge on the contents of Greenland seaweeds.

To address the lack of knowledge about the nutritional profile of Greenland seaweeds, the present study focused on ten seaweed species of interest, harvested in Greenland. The species chosen are either currently consumed in Greenland or Nordic countries (*Alaria esculenta* (Linnaeus) Greville 1830, *Ascophyllum nodosum* (Linnaeus) Le Jolis 1863, *Fucus vesiculosus* Linnaeus 1753, *Palmaria palmata* (Linnaeus) Weber & Mohr 1805, *Saccharina latissima* (Linnaeus) C.E. Lane & C. Mayes Druel & G.W. Saunders 2006, *Saccharina longicruris* (Bachelot de la Pylaie) Kuntze 1891), might conceivably be consumed (*Fucus distichus* Linnaeus 1767, *Hedophyllum nigripes* (J.Agardh) Starko & S.C.Lindstrom & Martone 2019, *Laminaria solidungula* J. Agardh 1868), or are potentially rich in bioactive components (*Agarum clathratum* Dumortier 1822). Since for some of the species, different parts of the thallus, the "body" of the macroalga, can constitute different products, they were analysed separately.

We hypothesise that the content of elements in Greenland seaweed species depends on species, thallus part and geographic origin. To study these hypotheses and to assess the suitability of these species as food items, the objectives of this study were to: (1) investigate the contents of beneficial and toxic elements in a range of Greenland seaweed species, (2) compare the element concentrations between different species, (3) investigate differences between thallus parts for selected species, (4) investigate the influence of geographic origin for *Fucus* spp., and (5) assess the benefits and limitations of the studied seaweeds as food items, through their contribution to recommended dietary intakes (RDIs), respectively the toxicological guideline values.

The findings from this study will be valuable for those currently collecting, farming, processing, marketing and consuming seaweeds in Greenland as well as the future development of the local seaweed sector.

## Materials and methods

### Samples and sampling locations

A total of 77 samples belonging to ten seaweed species were collected in the intertidal or upper subtidal zone between June and September in 2017 and 2018 at low tide conditions along the shore or by divers in West, South and East Greenland, see Fig 1. Table 1 provides an overview of the number of species per location. The harvest sites were chosen to represent different areas in Greenland. The species and number of samples were as following: *Agarum clathratum*

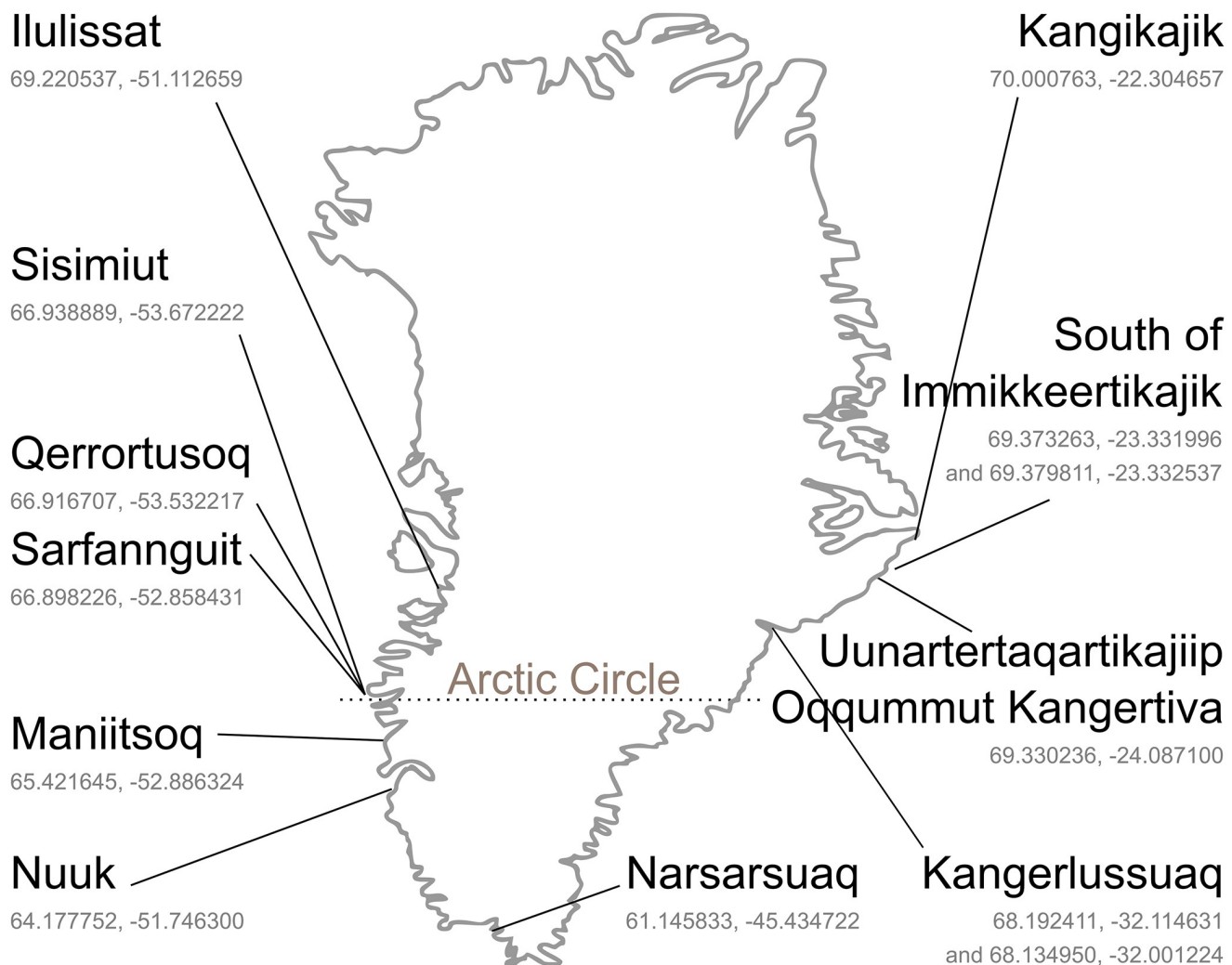

**Fig 1. Sampling locations in Greenland with coordinates in decimal degrees (latitude, longitude).** For Sarfannguit, coordinates are given for one central location (fish factory), specific coordinates for all three sampling sites are provided in Table 1.

(3), *Alaria esculenta* (9), *Ascophyllum nodosum* (7), *Fucus distichus* (8), *Fucus* spp. (7, specimens that were too small to be distinguished as either *F. distichus* or *F. vesiculosus*), *Fucus vesiculosus* (15), *Hedophyllum nigripes* (5), *Laminaria solidungula* (6), *Palmaria palmata* (2), *Saccharina latissima* (10) and *Saccharina longicruris* (3).

No permits were required for the described field study as none of the locations are privately owned or protected. This study did not involve endangered or protected species.

### Sample pre-treatment

Samples were rinsed in clean seawater at the collection site, epibiota were carefully removed, samples were frozen in clean food grade plastic bags at -20˚C and transported frozen to the laboratory in Denmark. Samples were freeze dried (Christ Beta 1-8, Martin Christ Gefriertrocknungsanlagen GmbH, Osterode am Harz, Germany) and, for compositional comparison of different thallus parts from five algal species, thereafter manually divided into blade, midrib and stipe, see Fig 2. Some of the received samples of *S. longicruris* and *S. latissima* had already

**Table 1. Summary of Greenland seaweed samples included in the study.** Coordinates in decimal degrees.

| Species | n | Location | Latitude | Longitude |
|---|---|---|---|---|
| *Agarum clathratum* | 2 | Kangerlussuaq | 68.134950 | -32.001224 |
| *Agarum clathratum* | 1 | Maniitsoq | 65.421645 | -52.886324 |
| *Alaria esculenta* | 3 | Kangerlussuaq | 68.134950 | -32.001224 |
| *Alaria esculenta* | 1 | Maniitsoq | 65.421645 | -52.886324 |
| *Alaria esculenta* | 1 | Nuuk | 64.177752 | -51.746300 |
| *Alaria esculenta* | 1 | Qerrortusoq | 66.916707 | -53.532217 |
| *Alaria esculenta* | 1 | South of Immikkeertikajik | 69.373263 | -23.331996 |
| *Alaria esculenta* | 2 | South of Immikkeertikajik | 69.379811 | -23.332537 |
| *Ascophyllum nodosum* | 1 | Maniitsoq | 65.421645 | -52.886324 |
| Ascophyllum nodosum | 3 | Narsarsuaq | 61.145833 | -45.434722 |
| Ascophyllum nodosum | 1 | Nuuk | 64.177752 | -51.746300 |
| Ascophyllum nodosum | 2 | Sisimiut | 66.938889 | -53.672222 |
| Ascophyllum nodosum | 1 | Sisimiut hospital | 66.943028 | -53.651677 |
| *Fucus distichus* | 4 | Kangerlussuaq | 68.192411 | -32.114631 |
| *Fucus distichus* | 1 | Maniitsoq | 65.421645 | -52.886324 |
| *Fucus distichus* | 1 | Nuuk | 64.177752 | -51.746300 |
| *Fucus distichus* | 2 | Sisimiut | 66.938889 | -53.672222 |
| *Fucus* spp. | 2 | Ilulissat kajak club | 69.220537 | -51.112659 |
| *Fucus* spp. | 3 | Sisimiut dump | 66.928316 | -53.673514 |
| *Fucus* spp. | 2 | Sisimiut hospital | 66.943028 | -53.651677 |
| *Fucus vesiculosus* | 1 | Maniitsoq | 65.421645 | -52.886324 |
| *Fucus vesiculosus* | 3 | Narsarsuaq | 61.145833 | -45.434722 |
| *Fucus vesiculosus* | 1 | Nuuk | 64.177752 | -51.746300 |
| *Fucus vesiculosus* | 1 | Qerrortusoq | 66.916707 | -53.532217 |
| *Fucus vesiculosus* | 2 | Sarfannguit dump | 66.897536 | -52.874138 |
| *Fucus vesiculosus* | 2 | Sarfannguit factory | 66.898226 | -52.858431 |
| *Fucus vesiculosus* | 2 | Sarfannguit school | 66.896311 | -52.857659 |
| *Fucus vesiculosus* | 3 | Sisimiut | 66.938889 | -53.672222 |
| *Fucus vesiculosus* | 1 | Sisimiut hospital | 66.943028 | -53.651677 |
| *Hedophyllum nigripes* | 1 | Kangikajik | 70.000763 | -22.304657 |
| *Hedophyllum nigripes* | 2 | Maniitsoq | 65.421645 | -52.886324 |
| *Hedophyllum nigripes* | 1 | Nuuk | 64.177752 | -51.746300 |
| *Hedophyllum nigripes* | 1 | South of Immikkeertikajik | 69.373263 | -23.331996 |
| *Laminaria solidungula* | 1 | Kangerlussuaq | 68.192411 | -32.114631 |
| *Laminaria solidungula* | 2 | South of Immikkeertikajik | 69.379811 | -23.332537 |
| *Laminaria solidungula* | 3 | Uunartertaqartikajiip Oqqummut Kangertiva | 69.330236 | -24.087100 |
| *Palmaria palmata* | 1 | Nuuk | 64.177752 | -51.746300 |
| *Palmaria palmata* | 1 | Sarfannguit factory | 66.898226 | -52.858431 |
| *Saccharina latissima* | 1 | Kangerlussuaq | 68.134950 | -32.001224 |
| *Saccharina latissima* | 3 | Kangikajik | 70.000763 | -22.304657 |
| *Saccharina latissima* | 5 | Sisimiut | 66.938889 | -53.672222 |
| *Saccharina latissima* | 2 | Uunartertaqartikajiip Oqqummut Kangertiva | 69.330236 | -24.087100 |
| *Saccharina longicruris* | 1 | Maniitsoq | 65.421645 | -52.886324 |
| *Saccharina longicruris* | 1 | Nuuk | 64.177752 | -51.746300 |

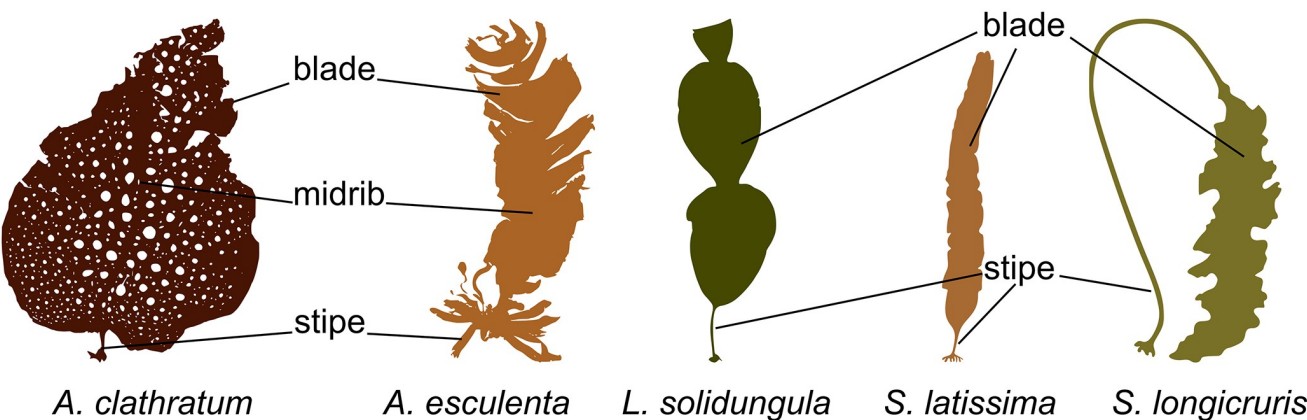

**Fig 2. Diagram showing the individually examined algal thallus parts of Greenland seaweeds:** *A. clathratum*, *A. esculenta*, *L. solidungula*, *S. latissima*, and *S. longicruris*.

been divided into stipe and blade. Additional epibiota was removed at this point. However, a limited presence of some epibiota, such as small crustaceans, especially on *A. nodosum*, cannot be ruled out due to the very branched structure of this macroalga. Homogenised powders were produced in a mill (Knifetec 1095 Sample Mill, FOSS, Hillerød, Denmark). Surplus sample material was saved for future studies.

## Analytical methods

All chemicals were of *pro analysi* quality or better, and all sample tubes were of inert quality to avoid contamination. All plastic tubes were new and all quartz digestion vessels were cleaned by microwave-assisted heating with concentrated nitric acid ($HNO_3$) (PlasmaPure, SCP Science, Courtaboeuf, France), and subsequent thorough rinsing with ultrapure water (18.2 MΩ at 25˚C, maximum 2 ppb total organic carbon, Milli-Q Integral 5 Water Purification System, Merck KGaA, Darmstadt, Germany).

This study was carried out using the principles in a modified and combined version of two reference methods, EN 13805:2014 [19] and EN 15763:2009 [20], for the determination of all elements except iodine.

An aliquot of 0.2 g seaweed powder was weighed into pre-weighed quartz digestion vessels to the nearest 1 mg. Two millilitre ultrapure water, followed by 4 mL concentrated $HNO_3$ were added to the sample. The samples were digested in a microwave reaction system (Multiwave 3000, Anton Paar GmbH, Graz, Austria). Following digestion, the samples were transferred to pre-weighed 50 mL centrifuge tubes, diluted to about 25 mL with ultrapure water and reweighed. Sample aliquots were further diluted (5, 100 and 1000 times) in 2% $HNO_3$ to a respective element concentration between 0 and 400 ng mL$^{-1}$ for analysis on the inductively coupled plasma mass spectrometry (ICP-MS) instrument. For each batch of 16 samples, two samples were determined in duplicates, one procedural blank (only ultrapure water and $HNO_3$) and one certified reference seaweed sample (NMIJ CRM 7405-a; Trace elements and arsenic compounds in seaweed (Hijiki, *Sargassum fusiforme*), National Metrology Institute of Japan, Tsukuba, Japan)) were processed and analysed alongside the rest of the samples. As internal standard, a mixture of Bi, In and Rh was prepared from single element calibration standards (PlasmaCal, SCP Science). A calibration curve from 0 to 400 ng mL$^{-1}$ was prepared for all elements from single element calibration standards (PlasmaCAL). The samples were

analysed on an 8900 ICP-MS Triple Quad (Agilent Technologies, Santa Clara, USA) equipped with an SPS4 Autosampler (Agilent Technologies).

For iodine, a modified version of the EN 15111:2007 [21] reference method was used. An aliquot of 0.3 g seaweed powder was weighed into pre-weighed 50 mL centrifuge tubes to the nearest 1 mg. Five millilitre of ultrapure water were added, and thoroughly mixed. Thereafter, 1 mL 25% tetramethylammonium hydroxide (TMAH) (TMAH, 25% w/w aq. soln., Alfa AesarTM, Fisher Scientific, Waltham, Massachusetts, USA) solution was added and the samples mixed again. The samples were heated in a drying oven '(Memmert UF 30, Memmert GmbH + Co. KG, Schwabach, Germany) at 90˚C for three hours. After 1.5 hours, the samples were removed briefly and inverted to ensure nothing stuck to the bottom. After cooling, the samples were diluted with 0.5% TMAH to approximately 50 mL and weighed again. A 5 mL aliquot of each sample was transferred to centrifuge tubes and centrifuged at 20 000 g for 5 min. The supernatant was then further diluted in 0.5% TMAH to a final iodine concentration between 0 and 100 ng mL$^{-1}$ for analysis on the ICP-MS instrument. Four procedural blanks and two certified replicate seaweed reference samples (3232 Kelp powder *Thallus laminariae*, NIST National Institute of Standards and Technology, Gathersburg US. ID 160129) were processed along with the samples. One in every ten samples was also determined in duplicate. Tellurium (Te) (PlasmaCal, SCP Science) was used as the internal standard and a calibration curve was prepared from ultrapure iodide (Iodide 1000 μg L$^{-1}$ Spectrascan SS11I, Ski, Norway) from 0 to 100 ng mL$^{-1}$ iodine. The samples were analysed on an iCAP Q ICP-MS (Thermo Scientific, Bremen, Germany) equipped with an ASX-520 AutoSampler (Cetac) running Qtegra version 2.10.3324.83 (64 bit) (Thermo Scientific).

The limit of detection (LOD) and limit of quantification (LOQ) for both methods were calculated using the standard deviation (SD) obtained from repeated analysis of blank samples:

LOD respectively LOQ = SD of the blanks (ng/mL) $^*$ f $^*$ dilution factor (mL)/sample amount (g), with f = 3 for LOD and f = 10 for LOQ.

The precision was determined from duplicate analysis of the same sample. The relative standard deviation (RSD), a measure of repeatability, was calculated from duplicate samples determined at different dates by the same operator on the same equipment.

ICP-MS parameters are summarised in Table A in S1 File.

For Na, I and P, LOD and LOQ were derived from the lowest accepted concentration of the calibration curve since the blanks were below detection limit. Relative standard deviations were in the range of 0.01% to 25%, with medians between 1.6% and 4.6% for all elements except Hg and Se with relative standard deviations up to 55% and 31%, respectively, and medians of 22% and 11%.

Assessment of normality revealed that element concentrations were not normally distributed. Therefore, element concentrations are reported as median values with median absolute deviation, and non-parametric tests were used for data analysis.

## Portion size survey

To determine an appropriate portion size for a seaweed salad served in Denmark, we asked six restaurants in the Copenhagen area and one Danish seaweed producer about their typical serving size for a seaweed salad. Only businesses that had seaweed salads on their menu or in their online store were contacted. Queries were sent by e-mail to the contact addresses available from the respective homepages. No personal data were collected and results were anonymised.

For the purpose of calculating the exposure to beneficial and toxic elements from eating a salad prepared from Greenland seaweed, we made the following assumptions: 1) such a salad would solely consist of one species of seaweed, 2) cutting up the fresh seaweed into a salad

would not change the observed concentrations and 3) dressing (if used) would constitute a negligible part of the total weight and nutritional content.

## Statistical analyses

For data analysis, results for those samples that had been divided into different thallus parts were pooled for analysis on species level. All data analysis was carried out with with R version 3.4.4 (2018-03-15) [22], using RStudio version 1.1.463 [23], on a x86_64-pc-linux-gnu (64-bit) platform. Data was imported via readxl [24], transformed using dplyr [25], analysed with the stats package, and visualised with ggplot2 [26]. For upload to external databases, data was exported with WriteXLS [27] or writecsv (utils package 3.4.4).

Normality was assessed with the Shapiro-Wilk test. Since preconditions for parametric tests were not met, Kendall's ranked correlation coefficient was used for pairwise element correlations and the Kruskal-Wallis test was used to compare differences between species and locations. Principal component analysis was carried out using the prcomp function (stats package 3.4.4), with centring and scaling of samples.

A confidence level of 95% was used unless otherwise noted.

## Results and discussion

### Quality assurance

Quality assurance parameters are presented in Table B in S1 File.

### Individual sample results

The full dataset of individual sample results is freely available online [28]. Samples with different thallus parts examined separately are connected by their sampleID to the respective powderIDs. In the present article, summarised results are presented and discussed.

### Species comparison

Median element contents are presented in Table 2 and Fig A in S2 File. The most abundant cations were K>Na>Ca>Mg (3.79 g kg$^{-1}$ to 108 g kg$^{-1}$ freeze dried weight). Schiener and colleagues [29] found the same sequence in an investigation that included among other seaweeds species, *S. latissima* and *A. esculenta*. These four light metal cations are also the most abundant cations in seawater, with Na>Mg>Ca>K [30]. In seaweeds, these cations are gradually replaced by heavier divalent metal ions such as Cu from the seawater [15], during the continued growth of the algae [31].

The next most abundant cation was Fe with 82.2 mg kg$^{-1}$ to 492 mg kg$^{-1}$, while the most abundant other elements were P with 1.00 g kg$^{-1}$ to 2.54 g kg$^{-1}$ and I with 113 mg kg$^{-1}$ to 4478 mg kg$^{-1}$. For all species, Hg concentrations were below the LOQ (0.078 mg kg$^{-1}$) or below the LOD (0.023 mg kg$^{-1}$). For most species, Se was below the LOQ (0.111 mg kg$^{-1}$), except for *A. clathratum*, *A. esculenta*, *F. distichus* and *Fucus* spp., which ranged from 0.142 mg kg$^{-1}$ to 0.227 mg kg$^{-1}$. All other elements (As, Cd, Cr, Cu, Mn, Ni, Pb, Zn) were in the range of 0.101 mg kg$^{-1}$ to 75.5 mg kg$^{-1}$.

*Palmaria palmata*, the only red seaweed studied, deviated mainly with respect to its lower content of As, Ca and I (6.93 mg kg$^{-1}$, 3.79 g kg$^{-1}$ and 113 mg kg$^{-1}$, respectively) compared to the brown seaweeds investigated. This is in accordance with previous studies [8–10, 32].

The high iodine contents (1466 mg kg$^{-1}$ up to 4478 mg kg$^{-1}$) found in Laminariaceae (*H. nigripes*, *L. solidungula*, *S. latissima* and *S. longicruris*) are in accordance with previous reports [8, 12, 32]. For *Laminaria digitata*, another member of the Laminariaceae family, Küpper and

**Table 2. Median content ± median absolute deviation of elements in Greenland seaweed samples, freeze dried weight.**

| Species | n | As (mg kg⁻¹) | Ca (g kg⁻¹) | Cd (mg kg⁻¹) | Cr (mg kg⁻¹) | Cu (mg kg⁻¹) | Fe (mg kg⁻¹) |
|---|---|---|---|---|---|---|---|
| *A. clathratum* | 3 | 46.1 ± 17.2 | 30.4 ± 7.1 | 0.208 ± 0.113 | 5.75 ± 1.18 | 4.51 ± 1.55 | 492 ± 368 |
| *A. esculenta* | 9 | 33.0 ± 14.8 | 18.2 ± 2.7 | 1.32 ± 0.66 | 7.04 ± 7.05 | 2.70 ± 1.54 | 306 ± 303 |
| *A. nodosum* | 8 | 29.8 ± 2.3 | 14.4 ± 2.5 | 0.293 ± 0.138 | 1.18 ± 0.11 | 5.65 ± 4.28 | 82.2 ± 25.8 |
| *F. distichus* | 8 | 40.1 ± 5.9 | 14.0 ± 1.0 | 0.952 ± 0.373 | 5.78 ± 5.85 | 19.1 ± 25.1 | 702 ± 834 |
| *Fucus* spp. | 7 | 26.6 ± 6.6 | 14.4 ± 1.8 | 0.826 ± 0.233 | 2.56 ± 1.17 | 4.32 ± 0.40 | 365 ± 342 |
| *F. vesiculosus* | 16 | 33.3 ± 7.6 | 13.4 ± 1.4 | 1.42 ± 1.01 | 1.29 ± 0.76 | 2.32 ± 1.43 | 119 ± 120 |
| *H. nigripes* | 5 | 63.1 ± 27.1 | 14.1 ± 3.0 | 0.168 ± 0.169 | 1.70 ± 1.54 | 2.85 ± 1.97 | 171 ± 111 |
| *L. solidungula* | 6 | 47.6 ± 13.1 | 10.6 ± 0.8 | 0.134 ± 0.036 | 1.61 ± 0.52 | 9.79 ± 10.25 | 406 ± 166 |
| *P. palmata* | 2 | 6.93 ± 1.97 | 3.79 ± 2.73 | 0.600 ± 0.271 | 0.657 ± 0.177 | 5.71 ± 3.57 | 131 ± 94 |
| *S. latissima* | 11 | 45.2 ± 12.1 | 10.8 ± 2.7 | 2.96 ± 1.08 | 1.89 ± 1.84 | 1.72 ± 1.33 | 124 ± 139 |
| *S. longicruris* | 2 | 61.9 ± 5.1 | 15.4 ± 1.3 | 1.25 ± 0.24 | 1.14 ± 0.58 | 1.40 ± 0.28 | 183 ± 169 |
| Species | n | Hg (mg kg⁻¹) | I (mg kg⁻¹) | K (g kg⁻¹) | Mg (g kg⁻¹) | Mn (mg kg⁻¹) | Na (g kg⁻¹) |
| *A. clathratum* | 3 | < 0.023* | 280 ± 46 | 37.5 ± 20.5 | 6.25 ± 0.65 | 13.5 ± 9.1 | 28.2 ± 2.1 |
| *A. esculenta* | 9 | < 0.078** | 502 ± 307 | 78.6 ± 31.3 | 9.20 ± 2.01 | 9.60 ± 8.30 | 48.2 ± 11.0 |
| *A. nodosum* | 8 | < 0.078** | 670 ± 162 | 20.5 ± 0.8 | 9.28 ± 1.42 | 11.0 ± 3.9 | 37.7 ± 4.5 |
| *F. distichus* | 8 | < 0.023* | 212 ± 43 | 33.1 ± 2.4 | 9.41 ± 1.26 | 36.5 ± 19.3 | 44.8 ± 4.6 |
| *Fucus* spp. | 7 | < 0.023* | 234 ± 44 | 36.7 ± 4.1 | 8.92 ± 1.05 | 26.8 ± 8.6 | 38.5 ± 5.2 |
| *F. vesiculosus* | 16 | < 0.023* ***** | 188 ± 74 | 25.5 ± 3.2 | 8.52 ± 0.97 | 34.3 ± 24.5 | 40.1 ± 6.6 |
| *H. nigripes* | 5 | < 0.078** | 3323 ± 742 | 90.6 ± 30.8 | 6.65 ± 0.55 | 4.69 ± 1.86 | 31.5 ± 3.9 |
| *L. solidungula* | 6 | < 0.078** | 4478 ± 1812 | 93.1 ± 29.6 | 4.92 ± 0.59 | 7.85 ± 5.05 | 24.5 ± 4.1 |
| *P. palmata* | 2 | < 0.023* | 113 ± 90 | 75.7 ± 18.2 | 4.63 ± 1.22 | 5.95 ± 2.85 | 35.0 ± 12.8 |
| *S. latissima* | 11 | < 0.078** | 3124 ± 927 | 59.1 ± 18.2 | 7.03 ± 2.07 | 4.18 ± 3.09 | 36.8 ± 10.1 |
| *S. longicruris* | 2 | < 0.023* | 1466 ± 702 | 108 ± 55 | 7.37 ± 1.24 | 3.35 ± 1.71 | 34.9 ± 3.4 |
| Species | n | Ni (mg kg⁻¹) | P (g kg⁻¹) | Pb (mg kg⁻¹) | Se (mg kg⁻¹) | Zn (mg kg⁻¹) | |
| *A. clathratum* | 3 | 3.35 ± 1.31 | 1.48 ± 0.59 | 0.337 ± 0.162 | 0.227 ± 0.089*** | 20.2 ± 12.7 | |
| *A. esculenta* | 9 | 3.52 ± 3.31 | 2.18 ± 1.42 | 0.474 ± 0.557 | 0.159 ± 0.044 | 19.7 ± 9.8 | |
| *A. nodosum* | 8 | 0.992 ± 0.133 | 1.06 ± 0.25 | 0.111 ± 0.110 | < 0.111** | 23.9 ± 11.4 | |
| *F. distichus* | 8 | 7.80 ± 1.42 | 1.26 ± 0.04 | 0.243 ± 0.066 | 0.142 ± 0.003**** | 17.4 ± 6.0 | |
| *Fucus* spp. | 7 | 5.99 ± 1.74 | 1.76 ± 0.37 | 1.59 ± 2.18 | 0.173 ± 0.090*** | 75.5 ± 37.9 | |
| *F. vesiculosus* | 16 | 3.33 ± 0.85 | 1.00 ± 0.33 | 0.101 ± 0.064 | < 0.111** ****** | 16.6 ± 12.0 | |
| *H. nigripes* | 5 | 1.40 ± 1.18 | 2.39 ± 0.39 | 0.158 ± 0.133 | < 0.111** | 30.9 ± 10.3 | |
| *L. solidungula* | 6 | 1.12 ± 0.63 | 2.04 ± 1.17 | 0.329 ± 0.266 | < 0.111** | 12.3 ± 4.5 | |
| *P. palmata* | 2 | 3.84 ± 2.28 | 2.54 ± 0.60 | 0.251 ± 0.274 | < 0.111** | 64.1 ± 56.0 | |
| *S. latissima* | 11 | 1.14 ± 0.85 | 2.24 ± 1.17 | 0.207 ± 0.218 | < 0.111** | 18.5 ± 15.2 | |
| *S. longicruris* | 2 | 0.783 ± 0.078 | 2.11 ± 0.62 | 0.641 ± 0.875 | < 0.111** | 21.3 ± 7.1 | |

* LOD,

** LOQ,

*** n = 2,

**** n = 4;

***** Hg was detected in a single sample of *F. vesiculosus* at levels below LOQ.

****** Se was detected in a single sample of of *F. vesiculosus* at 0.132 mg/kg dry weight.

colleagues [16] proposed that iodine, in the accumulated form of iodide, functions as an inorganic antioxidant. Ye and colleagues [17] also found iodoperoxidases in *Saccharina japonica*, another member of Laminariaceae. We therefore theorize that the four members of Laminariaceae studied here also possess iodoperoxidases, leading to the high observed accumulation of iodine.

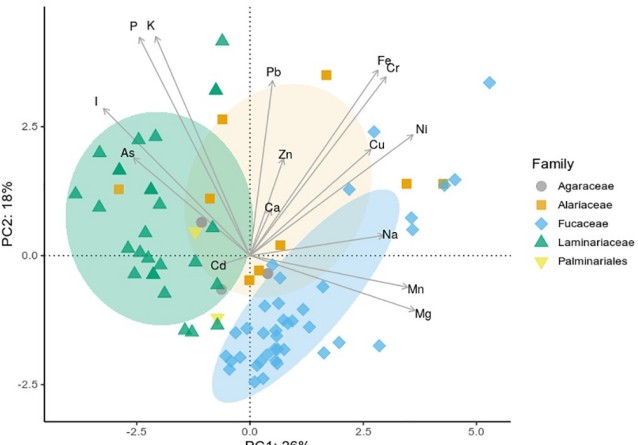

**Fig 3. Principal component analysis of element content on family level of Greenland seaweeds.** Agaraceae (*A. clathratum*), Alariaceae (*A. esculenta*), Fucaceae (*A. nodosum*, *F. distichus*, *F. vesiculosus* and *Fucus* spp.), Laminariaceae (*H. nigripes*, *L. solidungula*, *S. latissima* and *S. longicruris*) and Palminariales (*P. palmata*). Hg and Se were excluded from the analysis due to the low number of quantifiable samples. Ellipses denote 95% confidence intervals for Alariaceae, Fucaceae and Laminariaceae.

Riget, Johansen and Asmund [33] reported concentrations of selected elements (As, Cd, Cr, Cu, Fe, Pb, Zn) in *A. nodosum* and *F. vesiculosus* collected in the Nuuk area between 1980 and 1990. The major differences between their findings, and the findings of this study were increased Fe and Zn concentrations for both species. For *A. nodosum*, they reported Fe concentrations of 16 to 43 mg kg$^{-1}$, in this study we found 140 mg kg$^{-1}$, and Zn concentrations were reported as 6.6 to 10.7 mg kg$^{-1}$, while in this study we found 58.1 mg kg$^{-1}$. For *F. vesiculosus*, Fe concentrations of 33 to 77 mg kg$^{-1}$ were reported by Riget, Johansen and Asmund [33], while in this study we found 133 mg kg$^{-1}$ and they reported Zn concentrations of 7.2 to 10.2 mg kg$^{-1}$ compared to the 50.6 mg kg$^{-1}$ in this study. The most likely explanation could be the difference in sampling: Riget, Johansen and Asmund [33] collected five samples of growing tips, while in this study the entire thallus was analysed, and samples were pooled so this study only reports one result per species from Nuuk. Another explanation could be the increased human and industrial activity in the area since their study—Nuuk has nearly doubled in size, from around 9 000 inhabitants to close to 18 000 inhabitants [34]. This is supported by another study from Greenland investigating the influence of increased human activity [35] by monitoring Cr concentrations in indicator organisms, in this case mining. *Fucus vesiculosus* was used as one of the monitoring species, and Cr concentrations increased from 0.4 mg kg$^{-1}$ dry weight prior to mining operations, up to 2.62 mg kg$^{-1}$ dry weight during active operations of an open-pit mine in Southern Greenland. However, the Cr concentrations found in this study had not increased in a comparable manner to what was observed for the mining operation.

The presence of overall tendencies in element content or fingerprint per algal family were assessed by PCA, as presented in Fig 3. Both Hg and Se were excluded from the analysis due to the very low observed concentrations, which could not be quantified for the majority of samples. Fucaceae (*A. nodosum*, *F. distichus* and *F. vesiculosus*), characterised by a higher content of Mg, Mn and Ni, could clearly be distinguished from Laminariaceae (*H. nigripes*, *L. solidungula*, *S. latissima* and *S. longicruris*), which had a higher content of especially iodine, but also K and P. Alariaceae (*A. esculenta*) could not be distinguished from the other families with this method, and for Agaraceae (*A. clathratum*) and Palminariales (*P. palmata*), the low sample number of three and two samples, respectively, precluded analysis in this manner. To the best

of the authors knowledge, this is the first PCA presented in the literature of this specific combination of species. It is interesting to note that Laminariaceae, known for their high contents of iodine, could be distinguished from Fucaceae based mainly on their iodine content.

We also used PCA to investigate the influence of nearby human settlement size, based on the content of elements associated with anthropogenic contamination (Cd, Cr, Cu, Pb and Zn). However, there was no clear correlation evident.

## Thallus part comparison

For five species, a limited number of samples were divided into different parts, in particular blade, midrib and stipe: *A. clathratum* and *A. esculenta*, or blade and stipe: *L. solidungula*, *S. latissima* and *S. longicruris* (see Fig 2. for a schematic representation). Selected elements (As, Cd, Fe, I, K and Pb) are presented in Fig 4.

Concentrations of As were higher in stipes than in blades for *S. latissima* (Fig 4, panel A), and similarly, K concentrations in *S. latissima* and *S. longicruris* (Fig 4, panel E). A possible explanation for this could be that metal(loid)s (such as As, Cd, Hg, K and Pb) are stored associated with biopolymers [15], and these biopolymers are differently distributed throughout the thallus. Research into the properties of alginate, with respect to divalent metal ions, from *Laminara digitata* and *Laminaria hyperborea* in the 1960ies also showed differences between stipe and other (nondisclosed) parts of macroalgae [36] and is supported by observations by S. Wegeberg & O. Geertz-Hansen (unpublished data).

The comparison of reports on the concentrations (mg kg$^{-1}$) of total As in Icelandic *A. esculenta* from the Pétursdóttir and colleagues [37] and present studies showed for: stipes (53 ± 3 and 45 ± 6); midrib (43 ± 4 and 23 ± 7); and blade (93 ± 4 and 31 ± 16) respectively. With

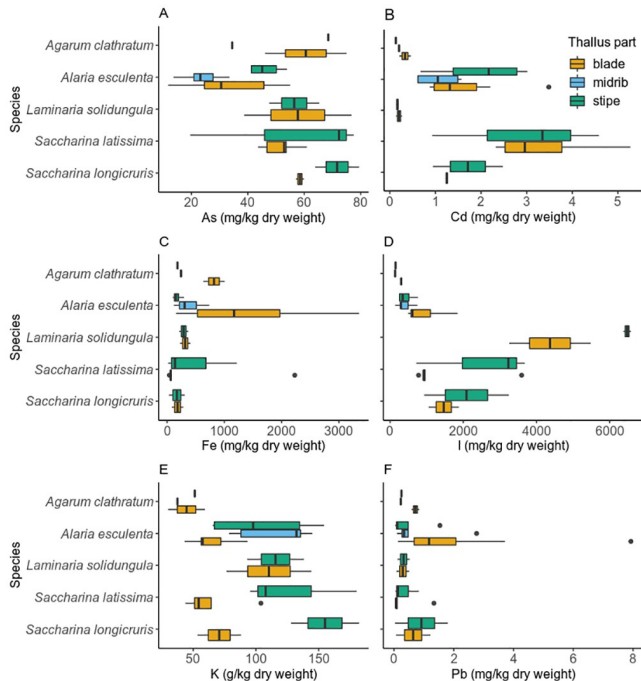

**Fig 4. Concentrations of elements (As, Cd, I, Fe, K and Pb) in different thallus parts of Greenland seaweeds for *A. clathratum*, *A. esculenta*, *L. solidungula*, *S. latissima* and *S. longicruris*.** The lower and upper hinges of the box represent the first and third quartile, around the median; the whiskers extend no further from the hinge than 1.5 * inter-quartile range. Outliers beyond the whiskers are shown as circles.

regards to the As content (mg kg$^{-1}$) in Icelandic *S. latissima*, the Pétursdóttir and colleagues [37] and present studies reported for: stipes (53 ± 4 and 75 ± 32); and blade ([117 ± 9–old frond and 116 ± 6–young frond] and 53 ± 7) respectively.

The differences between the study of Pétursdóttir and colleagues [37] and ours could be related to the time of sampling: they sampled during late winter, while the seaweeds in our study were collected during June to September. Previous studies have shown a seasonal change in nutritional composition of seaweeds (e.g. [29, 38]). Another possible explanation is the small sample size of both studies: two samples of Pétursdóttir and colleagues [37], whereas the present study reports on three stipe and five blade samples.

Ronan and colleagues [39] reported that total arsenic concentrations of both *A. nodosum* and *L. digitata* increased with the age of the thallus part, which is a probable explanation for the wide range of arsenic concentration observed in our study.

An explanation for the higher concentration of iodine in stipes compared to blades of *L. solidungula* and *S. latissima* (Fig 4, panel D) could be that these species possess iodoperoxidases, which are upregulated in parts of the macroalgae that are more exposed to environmental stress and pathogens, such as stipes, similarly to what Ye and colleagues [17] found for *S. japonica*. Another explanation is related to the age of the macroalgal part: while stipes are perennial, blades are annual.

Interestingly, some elements show a great variation in concentrations in the blade, such as Fe for *A. esculenta*, but not for any of the other Laminariaceae, see figure (Fig 4., panel C). This could be due to iron accumulating differently in older compared to younger macroalgae, or thallus parts, which have been shown to grow at different rates by Buggeln and colleagues [40].

Differences in element concentrations can also depend on where the sample is taken on the blade. This is due to the localization of meristem and thus the allocation of nutrients for growth, e.g., close to the stipe and hence close to the meristem or distally (S. Wegeberg and O. Geertz-Hansen (unpublished data on biopolymers), [29]). For *L. solidungula*, the blade generation is also significant. In the present study, neither the localisation on the blade nor the blade generation were investigated.

## Geographic origin comparison

All samples of *F. distichus* (n = 8), *F. vesiculosus* (n = 16) and *Fucus* spp. (n = 7) were used in a pooled investigation.

Fig 5 presents the results of the PCA, from which Hg and Se were again excluded, as previously for the species comparison. The samples can clearly be divided into Western (Maniitsoq, Nuuk, Qerrortusoq, Sarfannguit and Sisimiut), Southern (Narsarsuaq) and Eastern (Kangerlussuaq) origin. The three samples from the dump in Sisimiut illustrate the strong influence of human waste on the elemental composition of *Fucus* species. They are clearly separated along PC1 from the remaining Sisimiut samples, which include those from the hospital sewage outlet into Kangerluarsunnguaq, a bay with little water exchange. The samples from Ilulissat were collected within the city limits, where wastewater is diverted untreated into the sea. This human impact on the elemental composition is clearly reflected in the PCA, where the Ilulissat samples are grouped into the same quadrant as those from the dump in Sisimiut. By analysing many elements, it is thus possible to distinguish between locations even at small sample sizes per location. Analysis of location differences based on a single element through Kruskal-Wallis testing revealed statistically significant differences (p < 0.05) for the following elements: As, Cd, Cu, Fe, Mg, P, Pb and Zn. This is also reflected in the PCA, where these eight elements have the strongest influence on the PCs, as evidenced by the length of the arrows representing the loadings.

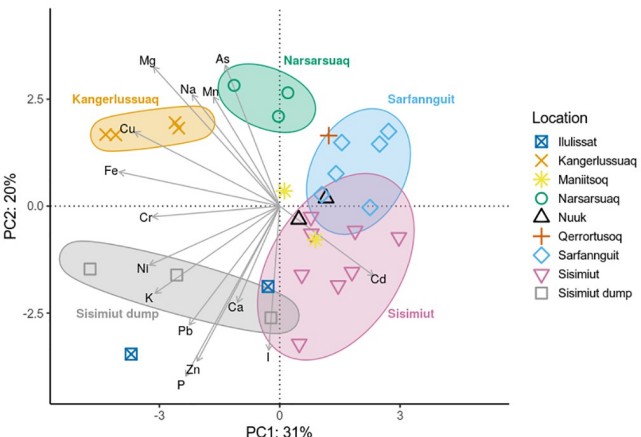

**Fig 5. Principal component analysis of element content of Greenland seaweeds depending on geographic location.**
All samples of *F. distichus*, *F. vesiculosus* and *Fucus* spp. were used in a pooled investigation. Hg and Se were excluded from the analysis due to the low number of quantifiable samples. Ellipses denote locations with at least three samples.

The observed natural variation at a given sampling site may be due to different factors, both abiotic (e.g. salinity) and biotic (e.g. fouling). These factors may lead to metabolic changes which affect growth rates and element uptake [41]. Brinza and colleagues [42] found that Zn uptake rates differed greatly between Danish and Irish *F. vesiculosus*. They explained the greater Zn uptake rates in the specimens from Denmark with differences in surface properties of the macroalgae, related to the salinity. At their collection site in the Sound, Denmark, salinity varies between 10 to 20 practical salinity units (PSU), compared to 36 PSU in Irish waters.

### Element correlations

Fig 6. summarises statistically significant values of Kendall's tau coefficient for pairwise element correlations. The strongest correlations were observed between Mg-Na (Kendall's tau

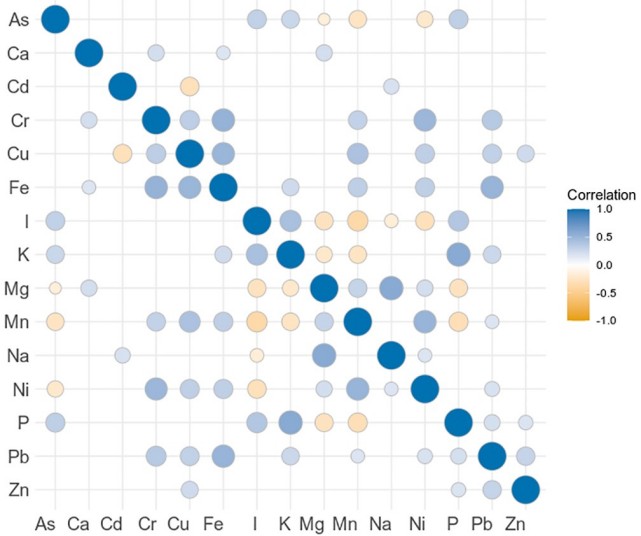

**Fig 6. Matrix of element correlations for Greenland seaweeds, expressed as Kendall's tau coefficient.** Elements are ordered alphabetically for ease of reading. Hg and Se were excluded from the analysis due to the low number of quantifiable samples. Only statistically significant correlations (p < 0.05) are shown.

0.58) and K-P (0.58), as well as Cr-Fe (0.52), Fe-Pb (0.51) and Ni-Mn (0.51). This is also reflected in the small angles between the loadings for K-P and Cr-Fe shown in Fig 3, a sign of correlation.

$Mg^{2+}$, $Na^+$ and $K^+$ are some of the most common cations in seawater [30, 43]. Seaweed acquires these light metal ions from seawater, and they are, together with $Ca^{2+}$, indeed reported as the main cations in seaweed biomass [15].

An explanation for the strong correlation observed between Cr and Fe can be that when Fe partially replaces Ca in the alginate matrix of the cell walls, it creates favourable binding sites for Cr, as reported by Nayak and colleagues [44].

A significant correlation between As-P (with a Kendall's tau coefficient of 0.32 in the present study), was also reported by Taylor and Jackson [45]. They reported a similar ratio of As-P in brown algae (0.015) as found in this study (0.025), which is slightly lower than the ratio found in seawater of 0.033 (As = 0.002 mg $L^{-1}$; P = 0.06 mg $L^{-1}$ [43]). They argued that this similarity in ratio could be due to these two elements being taken up by the same mechanism. However, we conclude that the difference in ratios between seawater and in the seaweed suggests that seaweed is indeed able to differentiate between As and P. Arsenates and phosphates have similar chemical properties, which contributes to the toxicity of arsenates [46]. In marine algae, As(V) enters cells through phosphate transporters, while As(III) enters through the plasma membrane via aquaglyceroporins and hexose permeases [46, 47]. Most of the arsenic taken up by macroalgae is stored as arsenosugars which are considered as less toxic to humans than inorganic arsenic [45].

Miedico and colleagues [48] also reported element correlations as Pearson's coefficients on 92 samples of edible seaweeds, and despite a significant overlap in the studied elements, only three element pairs were significantly correlated (p < 0.05) in both studies: As-Hg, Cr-Ni and Cu-Mn. These shared correlations are especially interesting since Miedico and colleagues [48] investigated different species, which furthermore originated from the Pacific. Thus these shared element pairs (As-Hg, Cr-Ni and Cu-Mn) indicate a relationship independent of macroalgae species and geographic origin.

This is supported by Desideri and colleagues [49], who reported element correlations as Pearson's coefficients on a total of 14 samples from a mixture of edible macroalgae and microalgae, some of the latter even being freshwater organisms. They did not indicate the threshold for significance, but element pairs significantly correlated in our study and those with high correlation coefficients in their study (> 0.7) were As-I, Cu-Mn and Ni-Mn.

We theorize that the shared correlations are due to chemical similarity of the element pairs, however an in-depth investigation is beyond the scope of this study.

## Nutritional and food safety aspects

Table 3 summarises current European and Nordic guidelines on recommended daily intake levels, upper daily intake levels, maximum levels in the EU and France and toxicological guideline values for the elements investigated in our study. Since average seaweed consumption data for Europe has not been documented, we based our intake scenario on a typical seaweed serving size in a Danish restaurant. From the four collected responses, the portion size of a seaweed salad ranged from 20 g to 50 g, with a median of 33 g. To assess the nutritional benefits and exposure to toxic elements, we calculated element concentrations found in a 33 g single-seaweed species salad, prepared from fresh seaweed (Table 4). Our estimated serving size is comparable with Sá Monteiro and colleagues [13], who estimated an intake of 5 g freeze dried weight per week, which corresponds to about 30 g fresh seaweed at an estimated moisture content of 80% (based on moisture contents reported by Holdt and Kraan [8]).

**Table 3. Current European and Nordic guidelines on recommended daily intake levels, upper daily intake levels, maximum levels in the EU and France and toxicological guideline values for the elements investigated.**

| Element | RI | UI | EU (mg kg$^{-1}$ ww) | France (mg kg$^{-1}$ dw) | Toxicological guideline value |
|---|---|---|---|---|---|
| As, inorganic | - | - | none | 3 [50] | 3 µg kg$^{-1}$ bw day$^{-1}$ BMDL$_{0.5}$ [51] |
| Ca | 800 mg [52] | 2.5 g [52] | - | - | - |
| Cd | - | - | 3.0* [53] | 0.5 [50] | 2.5 µg kg$^{-1}$ bw week$^{-1}$ TWI [54] |
| Cr | ** | ** | - | - | - |
| Cu | 0.9 mg [52] | 5 mg [52] | - | - | - |
| Fe | 9 mg or 14 mg*** [52] | 25 mg [52] | - | - | - |
| Hg | - | - | 0.10**** [55] | 0.1 [50] | 4 µg kg$^{-1}$ bw week$^{-1}$ inorganic Hg TWI, 1.3 µg kg$^{-1}$ bw week$^{-1}$ methylmercury TWI [56] |
| I | 150 µg [52, 57] | 600 µg [52, 57] | none | 2000 [50] | - |
| K | 4.7 g [58] | Low potassium diet: 2 g to 3 g [58] | - | - | - |
| Mg | 280 mg or 350 mg***** [52] | no recommendation | - | - | - |
| Mn | 3 mg [59] | no recommendation | - | - | - |
| Na | 575 mg; as salt 1.5 mg [52] | 2.4 g; as salt 6 g [52] | - | - | - |
| Ni | - | - | none | none | 2.8 µg kg$^{-1}$ bw day$^{-1}$ TDI [60] |
| P | 600 mg [52] | 3 g [52] | - | - | - |
| Pb | - | - | 3.0 **** [55] | 5 [50] | 0.50 µg kg$^{-1}$ bw day$^{-1}$ (developmental neurotoxicity) BMDL$_{0.1}$; 1.50 µg kg$^{-1}$ bw day$^{-1}$ (effects on systolic blood pressure) BMDL$_{0.1}$; 0.63 µg kg$^{-1}$ bw day$^{-1}$ (chronic kidney disease) BMDL$_{10}$ [61] |
| Se | 50 µg or 60 µg ***** [52] | 300 µg [52] | - | - | - |
| Zn | 7 mg or 9 mg ***** [52] | 25 mg [52] | - | - | - |

Abbreviations: Recommended daily intake (RI), upper daily intake (UI), lower confidence limit of the benchmark dose (BMDL), tolerable weekly intake (TWI), tolerable daily intake (TDI), body weight (bw), wet weight (ww), dry weight (dw).

The French regulations apply to seaweed in vegetable or condiment form.

* Food supplements consisting exclusively or mainly of dried seaweed, products derived from seaweed, or of dried bivalve molluscs.

** No recommendation given due to lack of sufficient evidence [62].

*** Lower value for men and women post menopause, higher value for women.

**** Food supplements.

***** Lower value for women and higher for men.

In general, all investigated seaweed species are good sources of essential minerals and trace elements. One portion of a single-seaweed salad contributes with between 1% to 55% of the recommended intake for a specific element. For example, one portion of *S. latissima* salad contains 647 µg Fe, corresponding to a daily recommended intake of 5% (for women) to 7% (for men and women post menopause).

However, iodine levels were high: *P. palmata* was the only seaweed for which iodine exposure did not exceed the recommended upper daily intake of 600 µg for adults, which is in accordance with other studies [10, 32]. However, it has been shown that iodine concentrations of e.g. *S. latissima* can greatly be reduced by soaking in warm freshwater [63] or blanching in hot freshwater [64]. A recent study in Ammassalik (East Greenland) by Andersen and colleagues [4] showed that consumption of locally harvested *A. nodosum* and *Chondrus crispus*

**Table 4. Calculated median element content for a single-seaweed salad of 33 g wet weight, prepared from Greenland seaweed.** Where applicable, percentage of recommended daily intake is indicated in parentheses. Elements exceeding recommended upper intake levels are marked in bold font.

| Species | As (µg) | Ca (mg) | Cd (µg) | Cr (µg) | Cu (µg) | Fe (µg) |
|---|---|---|---|---|---|---|
| *A. clathratum* | 281 | 185 (23) | 1.27 | 35.1 | 27.5 (3) | 3001 (33/21)* |
| *A. esculenta* | 147 | 81.1 (10) | 5.90 | 31.4 | 12.0 (1) | 1367 (15/10) |
| *A. nodosum* | 283 | 137 (17) | 2.79 | 11.1 | 53.7 (6) | 781 (10/6) |
| *F. distichus* | 283 | 98.8 (12) | 6.72 | 40.8 | 135 (15) | 4951 (55/35) |
| *F. vesiculosus* | 232 | 92.8 (12) | 9.84 | 8.97 | 16.1 (2) | 823 (10/6) |
| *H. nigripes* | 385 | 85.7 (11) | 1.02 | 10.4 | 17.4 (2) | 1043 (12/8) |
| *L. solidungula* | 254 | 56.6 (7) | 0.716 | 8.61 | 52.3 (6) | 2168 (24/16) |
| *P. palmata* | 34.9 | 19.1 (2) | 3.02 | 3.31 | 28.7 (3) | 660 (7/5) |
| *S. latissima* | 236 | 56.7 (7) | 15.5 | 9.86 | 9.01 (1) | 647 (7/5) |
| *S. longicruris* | 375 | 93.4 (12) | 7.57 | 6.93 | 8.50 (1) | 1113 (12/8) |
| Species | Hg (µg) | I (µg) | K (mg) | Mg (mg) | Mn (µg) | Na (mg) |
| *A. clathratum* | NA** | **1710** | 229 (5) | 38.1 (14/11)*** | 82.3 (3) | 172 (30) |
| *A. esculenta* | NA | **2243** | 351 (8) | 41.1 (15/12) | 42.9 (2) | 215 (37) |
| *A. nodosum* | NA | **6367** | 158 (4) | 88.2 (33/25) | 105 (3) | 358 (62) |
| *F. distichus* | NA | **1498** | 234 (5) | 66.4 (24/19) | 258 (9) | 316 (55) |
| *F. vesiculosus* | NA | **1305** | 178 (4) | 59.2 (21/17) | 238 (8) | 279 (49) |
| *H. nigripes* | NA | **20276** | 553 (12) | 40.6 (15/12) | 28.6 (1) | 192 (33) |
| *L. solidungula* | NA | **23901** | 497 (11) | 26.3 (9/8) | 41.9 (1) | 131 (23) |
| *P. palmata* | NA | 571 | 381 (8) | 23.3 (8/7) | 29.9 (1) | 176 (31) |
| *S. latissima* | NA | **16339** | 309 (7) | 36.8 (13/11) | 21.8 (1) | 192 (33) |
| *S. longicruris* | NA | **8897** | 655 (14) | 44.7 (16/13) | 20.3 (1) | 211 (37) |
| Species | Ni (µg) | P (mg) | Pb (µg) | Se (µg) | Zn (µg) | |
| *A. clathratum* | 20.4 | 9.01 (2) | 2.05 | 1.389 (3/2)*** | 124 (2/1)*** | |
| *A. esculenta* | 15.7 | 9.75 (2) | 2.12 | 0.710 (1/1) | 87.8 (1/1) | |
| *A. nodosum* | 9.43 | 10.0 (2) | 1.06 | NA | 227 (3/2) | |
| *F. distichus* | 55.0 | 8.92 (2) | 1.71 | 0.999 (2/2) | 123 (2/1) | |
| *F. vesiculosus* | 23.1 | 6.98 (1) | 0.701 | NA | 115 (2/1) | |
| *H. nigripes* | 8.51 | 14.6 (2) | 0.964 | NA | 189 (3/2) | |
| *L. solidungula* | 5.96 | 10.9 (2) | 1.76 | NA | 65.4 (1/1) | |
| *P. palmata* | 19.3 | 12.8 (2) | 1.26 | NA | 323 (5/4) | |
| *S. latissima* | 5.96 | 11.7 (2) | 1.08 | NA | 96.7 (1/1) | |
| *S. longicruris* | 4.75 | 12.8 (2) | 3.89 | NA | 130 (2/1) | |

* Lower value for men and women post menopause, higher value for women.

** Concentration below limit of quantification, see also Table 2.

*** Lower value for women and higher for men.

led to elevated urinary iodine excretion. After the ingestion of a 45 g seaweed meal, iodine was reported to be excreted after 36 hours [4]. The iodine richer *A. nodosum* led to higher excretion values, but overall bioavailability was about 50%. They also found that iodine excretion levels correlated to the reported frequency of seaweed consumption, with higher excretion levels for individuals reporting frequent intake of seaweed. Another study carried out in Nuuk, West Greenland, by Noahsen and colleagues [65] found that the consumption of a sushi meal comprised of a halibut maki roll with a 25 g *F. vesiculosus* salad led to increased urinary iodine excretion and elevated serum thyroid stimulating hormone (TSH), while no effect on serum estimated-free thyroxine (also known as T4) was observed. Urinary iodine excretion returned

to pre-meal levels by day 2 post-meal, and TSH by day 3. They concluded that a single meal containing seaweed only had a temporary effect on the thyroid, even at high iodine concentrations in the food.

Furthermore, it is important to note that, while K is an important constituent of the human diet, for patients on a low potassium diet (2 to 3 g day$^{-1}$), the consumption of one seaweed salad prepared from *H. nigripes*, *L. solidungula* or *S. longicruris* would contribute with over 0.5 g K, which is up to 25% of the recommended daily intake for these patients.

None of the individual samples exceeded the EU maximum levels for Hg of 3 mg kg$^{-1}$ wet weight, see also Table 3, for individual sample results see [28]. Many samples (48) exceeded the maximum levels for Cd according to French regulations. However, the French Agency for Food, Environmental and Occupational Health & Safety (ANSES) is currently evaluating whether these maximum levels will be maintained or increased to the European level, which was not exceeded by any sample. Only two samples exceeded the French limit for Pb, but none exceeded the European limit for Pb. Many samples of Laminariaceae (*H. nigripes*, *L. solidungula*, *S. latissima*, *S. longicruris*) exceeded the French regulation maximum level for iodine.

The content of total arsenic is listed for future reference, since the content of inorganic arsenic, for which there exist toxicological guideline values, was not quantified in this study.

## Conclusion

In this study, 77 samples of ten Greenland seaweed species were collected and analysed for the content of 17 elements.

The element profiles varied between species, and species from the same family tended to have similar profiles. For those species where different parts of the thallus were investigated, the element concentrations varied between different parts, in accordance with other studies. The results from the thallus part analysis of stipe, rib or blade can be used to select or discard specific seaweed parts, depending on desired high or low concentrations of specific elements.

Elements associated with anthropogenic contamination showed no clear trend with human settlement size. Broad geographic differentiation, based on element profile, was possible for *Fucus* species. However, the geographic identification was obfuscated in the case of Sisimiut and Ilulissat, for samples collected close to waste discharge. The strong influence of human waste on the elemental profile means one should refrain from harvesting close and downstream to waste discharge into the sea, even though current European limits for toxic elements were not exceeded.

Iodine contents were very high in some species of the Laminariaceae, which limits consumption of untreated raw macroalgae according to recommendations on daily intake. However, studies on washing and blanching treatments of seaweeds from other areas show that these treatments are very effective in iodine reduction, while maintaining a good nutritional profile. Recent studies in Greenland furthermore suggest that bioavailability of iodine from seaweed might be as low as 50%, and that intake of a single meal containing seaweed only had a temporary effect on the thyroid.

Future studies should focus on the influence of post-harvest treatments prior to consumption, such as drying, blanching or fermentation, on the nutritional profile of seaweeds from Greenland.

Furthermore, a more detailed investigation of seaweeds from different areas and substratum will help to elucidate geographic differences.

## Supporting information

**S1 File.**
(XLSX)

**S2 File.**
(PDF)

## Acknowledgments

We are grateful for the collection of part of the samples in Sisimiut to Ulrik Lyberth. Birgitte Koch Herbst and Annette Landin are thanked for invaluable help with, and training in the laboratory analyses.

## Author Contributions

**Conceptualization:** Katharina J. Kreissig, Lisbeth Truelstrup Hansen, Pernille Erland Jensen.

**Data curation:** Katharina J. Kreissig.

**Formal analysis:** Katharina J. Kreissig, Jens J. Sloth.

**Funding acquisition:** Katharina J. Kreissig, Lisbeth Truelstrup Hansen, Pernille Erland Jensen, Susse Wegeberg, Ole Geertz-Hansen.

**Investigation:** Katharina J. Kreissig.

**Project administration:** Katharina J. Kreissig, Lisbeth Truelstrup Hansen, Pernille Erland Jensen.

**Resources:** Lisbeth Truelstrup Hansen, Susse Wegeberg, Ole Geertz-Hansen, Jens J. Sloth.

**Supervision:** Lisbeth Truelstrup Hansen, Pernille Erland Jensen, Susse Wegeberg, Ole Geertz-Hansen, Jens J. Sloth.

**Visualization:** Katharina J. Kreissig.

**Writing – original draft:** Katharina J. Kreissig.

**Writing – review & editing:** Lisbeth Truelstrup Hansen, Pernille Erland Jensen, Susse Wegeberg, Ole Geertz-Hansen, Jens J. Sloth.

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
