## [Decision Letter · Decision Letter 0]

31 Dec 2020

PONE-D-20-36449

Characterisation and chemometric evaluation of 17 elements in ten seaweed species from Greenland

PLOS ONE

Dear Dr. Kreissig,

Thank you for submitting your manuscript to PLOS ONE. After careful consideration, we feel that it has merit but does not fully meet PLOS ONE’s publication criteria as it currently stands. Therefore, we invite you to submit a revised version of the manuscript that addresses the points raised during the review process.

We look forward to receiving your revised manuscript.

Kind regards,

Yi Hu

Academic Editor

PLOS ONE

Journal Requirements:

2.) Please provide additional details regarding participant consent to collect personal data, including email addresses, names, or phone numbers. In the Methods section, please ensure that you have specified how consent was obtained and how the study met relevant personal data and privacy laws. If data were collected anonymously, please include this information.

3.) We note that Figure 1 in your submission contains map images which may be copyrighted.

We require you to either (a) present written permission from the copyright holder to publish this figure specifically under the CC BY 4.0 license, or (b) remove the figures from your submission:

a.) You may seek permission from the original copyright holder of Figure 1 to publish the content specifically under the CC BY 4.0 license. 

b.) If you are unable to obtain permission from the original copyright holder to publish this figure under the CC BY 4.0 license or if the copyright holder’s requirements are incompatible with the CC BY 4.0 license, please either i) remove the figure or ii) supply a replacement figure that complies with the CC BY 4.0 license. Please check copyright information on all replacement figures and update the figure caption with source information. If applicable, please specify in the figure caption text when a figure is similar but not identical to the original image and is therefore for illustrative purposes only.

Reviewers' comments:

Reviewer's Responses to Questions

**Comments to the Author**

1. Is the manuscript technically sound, and do the data support the conclusions?

Reviewer #1: Yes

Reviewer #2: Yes

2. Has the statistical analysis been performed appropriately and rigorously? 

Reviewer #1: Yes

Reviewer #2: Yes

3. Have the authors made all data underlying the findings in their manuscript fully available?

Reviewer #1: Yes

Reviewer #2: Yes

4. Is the manuscript presented in an intelligible fashion and written in standard English?

Reviewer #1: Yes

Reviewer #2: Yes

5. Review Comments to the Author

Reviewer #1: Manuscript Number: PONE-D-20-36449

Title: Characterisation and chemometric evaluation of 17 elements in ten seaweed species from Greenland.

The subject of this manuscript falls within the general scope of Plos One. The manuscript is an original contribution to the field of using seaweeds as products with nutraceutical interest. The interpretations and conclusions are justified by the results. I can recommend the publication of this manuscript in Plos One after major revision.

Keywords are missing?

Introduction

Lines 32-36. Please, add the name of the authors to the scientific names of the species the first time you cite them.

Line 44. Please, add your hypothesis at the end of the Introduction section. May be you want to extend your Introduction a bit to justify your hypotheses.

Materials and methods.

Lines 138-139. Please, provide the name of the restaurants and the sea weed producer in supplemental material.

Results

Lines 162-166. This paragraph should go in Material and Methods.

Lines 167-169. This paragraph should go in Material and Methods (Statistical analyses).

Lines 310-311. You haven’t proved this to say: ‘The observed natural variation at a given sampling site is due to different factors, both abiotic (e.g. salinity) and biotic (e.g. fouling)’. You should always write in conditional tense if you have not proved what you are commenting in this sentence and although the text.

I think it would be clearer if Results would be shown separately from the Discussion

Tables

Tables 2 and 4 should be converted into one or more figures including different histograms, where it would be easier to see the differences between species (as you have done in Figure 4. Actually Figure 4 seems redundant with Table 2?). May be Table 4 could be converted in a stacked bar or a circle diagram.

Table 4. Please indicate how many grams of seaweeds would include the salad.

I have not properly corrected the English writing since it is not my native language.

Reviewer #2: The manuscript is fine, and should be accepted after effecting the attached review comments.

6. PLOS authors have the option to publish the peer review history of their article (what does this mean?). If published, this will include your full peer review and any attached files.

Reviewer #1: No

Reviewer #2: **Yes: **Dr. Millicent Uzoamaka Ibezim-Ezeani

---

## [Author Response · Author response to Decision Letter 0]

21 Jan 2021

We would like to thank the reviewers for taking the time to review our manuscript carefully. We have taken great care to respond to each of the comments and believe the manuscript has been improved. Our answers to each of the raised points are indicated by >. 

Academic editor 

1.) PLOS ONE's style requirements 

> We have followed PLOS ONE’s style requirements exactly, using the LaTeX template provided by PLOS ONE and checking all figures through PACE. 

2.) Details regarding participant consent 

> Businesses were contacted through their official email addresses available from their homepages, and we collected no personal data. The identity of the businesses and the results were anonymised. We have clarified this in the methods section. 

3.) Figure 1 – possible copyright of map images 

> Katharina J. Kreissig created the illustration in Inkscape. The map outline used is modified from the public domain figure “Arctic-umiaq-line-ports-of-call.svg” - downloaded from Wikimedia commons here: 

https://commons.wikimedia.org/wiki/File:Arctic-umiaq-line-ports-of-call.svg (Last accessed 2021-01-14). We have included a separate statement about the figure with our resubmission. 

Reviewer #1: 

Keywords are missing? 

> We would like to refer to page 1 of the compiled first submission “PONE-S-20-45496.pdf”, where keywords are listed. 

Introduction 

Lines 32-36. Please, add the name of the authors to the scientific names of the species the first time you cite them. 

> We have incorporated this suggestion into our revision. 

Line 44. Please, add your hypothesis at the end of the Introduction section. May be you want to extend your Introduction a bit to justify your hypotheses. 

> We have incorporated this suggestion into our revision. 

Materials and methods. 

Lines 138-139. Please, provide the name of the restaurants and the seaweed producer in supplemental material. 

> We have intentionally anonymised the results. The materials and methods section has been adjusted to reflect this. No personal information was collected, and the identity of the restaurants/producer had no impact on the result (i.e., average portion size of seaweed salad). 

Results 

Lines 162-166. This paragraph should go in Material and Methods. 

> We have incorporated this suggestion into our revision. 

Lines 167-169. This paragraph should go in Material and Methods (Statistical analyses). 

> We have incorporated this suggestion into our revision. 

Lines 310-311. You haven’t proved this to say: ‘The observed natural variation at a given sampling site is due to different factors, both abiotic (e.g. salinity) and biotic (e.g. fouling)’. You should always write in conditional tense if you have not proved what you are commenting in this sentence and although the text. 

> We have incorporated this suggestion into our revision. 

I think it would be clearer if Results would be shown separately from the Discussion 

> We respectfully disagree. We believe that for this article, it is better to have a combined results and discussion section, as there otherwise would be extensive repetition of element concentrations. Also, we respectfully refer to the author guideline for the journal, which permits the use of a combined results and discussion section. 

Tables 

Tables 2 and 4 should be converted into one or more figures including different histograms, where it would be easier to see the differences between species (as you have done in Figure 4. Actually Figure 4 seems redundant with Table 2?). May be Table 4 could be converted in a stacked bar or a circle diagram. 

> Table 2 is a central result of the study. Readers of the article will be interested in these exact numbers to compare with results from their own studies. 

> However, to also accommodate the reviewer’s great suggestion and as a service to the readers we have now included boxplots in the supplementary materials figure A to provide a graphical overview of the results presented in Table 2. 

> Figure 4 shows more detailed information about different thallus parts for selected seaweed species. This information is not available in Table 2, which shows median contents of the entire macroalgae. 

> Table 4 is very information dense, and we do not feel we can represent it appropriately as a figure without loss of overview of the data. 

Table 4. Please indicate how many grams of seaweeds would include the salad. 

> We have incorporated this suggestion into our revision. 

I have not properly corrected the English writing since it is not my native language. 

Reviewer #2 

> We have incorporated all of your suggestions into our revision. They were very helpful. Thank you for your help.

---

## [Decision Letter · Decision Letter 1]

25 Jan 2021

Characterisation and chemometric evaluation of 17 elements in ten seaweed species from Greenland

PONE-D-20-36449R1

Dear Dr. Kreissig,

We’re pleased to inform you that your manuscript has been judged scientifically suitable for publication and will be formally accepted for publication once it meets all outstanding technical requirements.

Kind regards,

Yi Hu

Academic Editor

PLOS ONE

Additional Editor Comments (optional):

Reviewers' comments:

Reviewer's Responses to Questions

**Comments to the Author**

1. If the authors have adequately addressed your comments raised in a previous round of review and you feel that this manuscript is now acceptable for publication, you may indicate that here to bypass the “Comments to the Author” section, enter your conflict of interest statement in the “Confidential to Editor” section, and submit your "Accept" recommendation.

Reviewer #1: All comments have been addressed

2. Is the manuscript technically sound, and do the data support the conclusions?

Reviewer #1: Yes

3. Has the statistical analysis been performed appropriately and rigorously? 

Reviewer #1: Yes

4. Have the authors made all data underlying the findings in their manuscript fully available?

Reviewer #1: Yes

5. Is the manuscript presented in an intelligible fashion and written in standard English?

Reviewer #1: Yes

6. Review Comments to the Author

Reviewer #1: All my previous comments have been addressed. I have not properly corrected the English writing since it is not my native language.

7. PLOS authors have the option to publish the peer review history of their article (what does this mean?). If published, this will include your full peer review and any attached files.

Reviewer #1: No

---

## [Editor Report · Acceptance letter]

28 Jan 2021

PONE-D-20-36449R1 

Characterisation and chemometric evaluation of 17 elements in ten seaweed species from Greenland 

Dear Dr. Kreissig:

I'm pleased to inform you that your manuscript has been deemed suitable for publication in PLOS ONE. Congratulations! Your manuscript is now with our production department. 

Kind regards, 

on behalf of

Prof. Yi Hu 

Academic Editor

PLOS ONE